

# Model-Based Clustering and Variable Selection for Multivariate Count Data

COMPUTO

ISSN 2824-7795

Julien Jacques[1]    Laboratoire ERIC, Université de Lyon

Thomas Brendan Murphy    School of Mathematics & Statistics, University College Dublin

Institut d'Études Avancées, Université de Lyon

Date published: 2025-03-27    Last modified: 2025-03-27

## Abstract

Model-based clustering provides a principled way of developing clustering methods. We develop a new model-based clustering methods for count data. The method combines clustering and variable selection for improved clustering. The method is based on conditionally independent Poisson mixture models and Poisson generalized linear models. The method is demonstrated on simulated data and data from an ultra running race, where the method yields excellent clustering and variable selection performance.

*Keywords:* Count data, Model-based clustering, Variable selection

# Contents

[1]Corresponding author: julien.jacques@univ-lyon2.fr

# 1 Introduction

Multivariate count data is ubiquitous in statistical applications, as ecology (Chiquet, Mariadassou, and Robin 2021), genomics (Rau et al. 2015; Silva et al. 2019). These data arise when each observation consists of a vector of count values. Count data are often treated as continuous data and therefore modeled by a Gaussian distribution, this assumption is particularly poor when the measured counts are low. Instead, we use the reference distribution for count data which is the Poisson distribution (Agresti 2013; Inouye et al. 2017a).

When a data set is heterogeneous, clustering allows to extract homogeneous subsets from the whole data set. Many clustering methods, such as $k$-means (Hartigan and Wong 1979), are geometric in nature, whereas many modern clustering approaches are based on probabilistic models. In this work, we use model-based clustering which has been developed for many types of data (Bouveyron et al. 2019; McLachlan and Peel 2000; Frühwirth-Schnatter, Celeux, and Robert 2018).

Modern data are often high-dimensional, that is the number of variables is often large. Among these variables, some are useful for the task of interest, some are useless for the task of interest and some others are useful but redundant. There is a need to select only the relevant variables, and that whatever is the task. Variable selection methods are widespread for supervised learning tasks, in particular to avoid overfitting. However, variable selection methods are less well developed for unsupervised learning tasks, such as clustering. Recently, several methods have been proposed for selecting the relevant variables in model-based clustering; we refer to Fop and Murphy (2018) and McParland and Murphy (2018) for recent detailed surveys.

The goal of the present work is to provide a clustering and variable selection method for multivariate count data, which, to the best of our knowledge, has not yet been studied in depth. A methodology based on a conditionally independent Poisson mixture is developed to achieve this goal. The method yields a final clustering model which is a conditionally independent Poisson mixture model for a subset of the variables.

# 2 Motivating Example

The International Association of Ultrarunners (IAU) 24 hour World Championships were held in Katowice, Poland from September 8th to 9th, 2012. Two hundred and sixty athletes representing twenty four countries entered the race, which was held on a course consisting of a 1.554 km looped route. An update of the number of laps covered by each athlete was recorded approximately every hour (White and Murphy 2016). Figure 1 plots the number of loops recorded each hour for the three medalists.

We can see among these three runners different strategies, the second placed runner lapped at a regular rate, the first placed runner had a fast start but slowed later, and the third placed runner also started fast but slowed more than the first place runner.

Our first goal will be, to analyze the whole data set to identify the different running strategies and to evaluate which strategies are the best ones. The second goal is to identify which variables allows to distinguish between the clusters, in order to identify which hour is essential in the management of this endurance race.

# 3 Independent Poisson Mixture

Let $X_n = (X_{n1}, X_{n2}, \ldots, X_{nM})$ be a random vector of counts for $n = 1, 2, \ldots, N$. The goal is to clusters theses $N$ observations into $G$ clusters. Let $Z_n = (Z_{n1}, Z_{n2}, \ldots, Z_{nG})$ be the latent cluster indicator

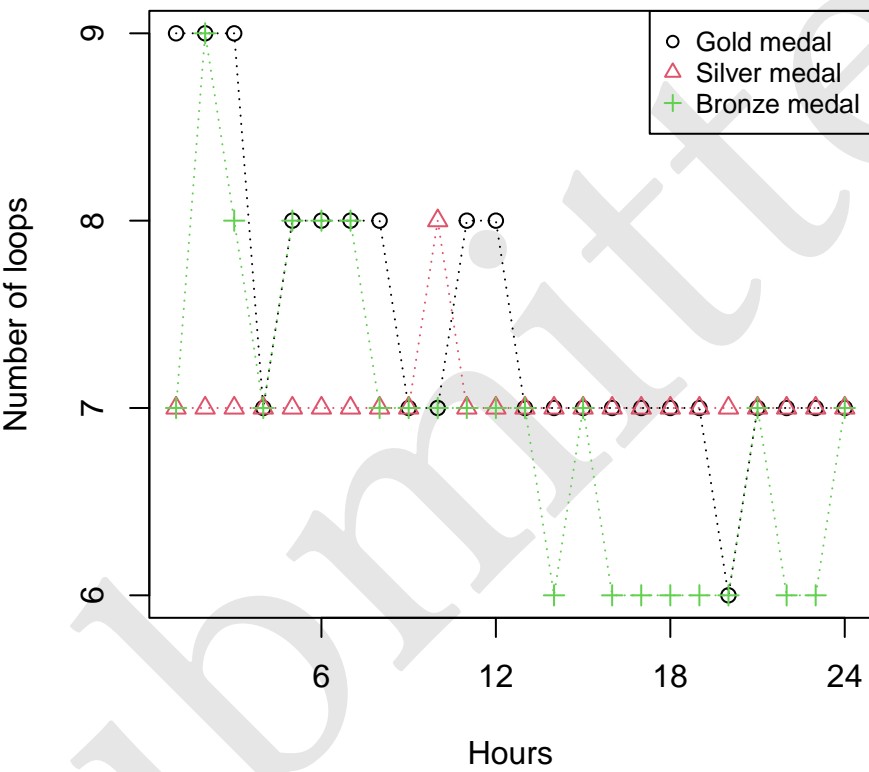

Figure 1: Number of loops per hour for the three medalists.

60 vector, where $Z_{ng} = 1$ if observation $n$ belongs to cluster $g$ and $Z_{ng} = 0$ otherwise. We assume that
61 $\mathbb{P}\{Z_{ng} = 1\} = \tau_g$ for $g = 1, 2, \ldots, G$. Let denote $\tau = (\tau_1, \ldots, \tau_G)$. The conditionally independent Poisson
62 mixture model (Karlis 2018, sec. 9.4.2.1) assumes that the elements of $X_n$ are independent Poisson
63 distributed random variables, conditional on $Z_n$. That is,

$$Z_n \sim \text{Multinomial}(1, \tau)$$
$$X_{nm}|(Z_{ng} = 1) \sim \text{Poisson}(\lambda_{gm}), \text{ for } m = 1, 2, \ldots, M.$$

64 Alternative modelling frameworks exist, either to introduce some dependence between variables
65 or to normalize the variables. We refer the interested reader to (Karlis 2018; Bouveyron et al. 2019,
66 chap. 6) for more details.

67 Denoting the model parameters by $\theta = (\tau, \lambda)$ where $\lambda = (\lambda_{gm})_{1 \leq g \leq G, 1 \leq m \leq M}$, and where $X = (x_n)_{1 \leq n \leq N}$
68 denotes the observations, the observed likelihood is

$$L(\theta) = \sum_{n=1}^{N} \sum_{g=1}^{G} \tau_g \prod_{m=1}^{M} \phi(x_{nm}, \lambda_{gm}),$$

69 where $\phi(x, \lambda) = \exp(-\lambda)\lambda^x/x!$, the Poisson probability mass function.

70 Due to form of the mixture distribution, there are no closed form for the maximum likelihood
71 estimators, and an iterative EM algorithm needs to be used (Dempster, Laird, and Rubin 1977) to
72 maximize the likelihood. The EM algorithm consists, starts from an initial value $\theta^{(0)}$ for the model
73 parameter, and alternates the two following steps until convergence of the likelihood.

74 At the $q$th iteration of the EM algorithm, the E-step consists of computing for all $1 \leq n \leq N$ and
75 $1 \leq g \leq G$:

$$t_{ng}^{(q)} = \frac{\tau_g^{(q)} \prod_{m=1}^{M} \phi(x_{nm}, \lambda_{gm})}{\sum_{h=1}^{G} \tau_h^{(q)} \prod_{m=1}^{M} \phi(x_{nm}, \lambda_{hm})}.$$

76 In the M-step, the model parameters are updated as follows:

$$\tau_g^{(q+1)} = \frac{\sum_{n=1}^{N} t_{ng}^{(q)}}{N} \quad \text{and} \quad \lambda_{gm}^{(q+1)} = \frac{\sum_{n=1}^{N} t_{ng}^{(q)} x_{nm}}{\sum_{n=1}^{N} t_{ng}^{(q)}}.$$

77 The EM algorithm steps are iterated until convergence, where convergence is determined when
78 $\log L(\theta^{(q+1)}) - \log L(\theta^{(q)}) < \epsilon$.

79 The number of clusters $G$ is selected using the Bayesian information criterion (BIC) (Schwarz 1978),

$$BIC = 2 \log L(\hat{\theta}) - \{(G - 1) + GM\} \log(N),$$

80 where $\hat{\theta}$ is the maximum likelihood estimate of the model parameters; models with higher BIC are
81 prefered to models with lower BIC.

## 4  Variable selection

83 We develop a model-based clustering method with variable selection for multivariate count data.
84 The method follows the approach of (Raftery and Dean 2006; Maugis, Celeux, and Martin-Magniette
85 2009) for continuous data and (Dean and Raftery 2010; Fop, Smart, and Murphy 2017) for categorical
86 data. It consists in a stepwise model comparison approach where variables are added and removed
87 from a set of clustering variables.

### 4.1 Model setup

The clustering and variable selection approach is based around partitioning $X_n = (X_n^C, X_n^P, X_n^O)$ into three parts:

- $X_n^C$: The current clustering variables,
- $X_n^P$: The proposed variable to add to the clustering variables,
- $X_n^O$: The other variables.

For simplicity of notation, $C$ will be used to denote the set of indices of the current clustering variables, $P$ the indices of the proposed variable and $O$ the indices of the other one. Then $(C, P, O)$ is a partition of $\{1, \ldots, M\}$.

The decision on whether to add the proposed variable to the clustering variables is based on comparing two models:

$M_1$ (Clustering Model), which assumes that the proposed variable is useful for clustering:

$$(X_n^C, X_n^P) \sim \sum_{g=1}^{G} \tau_g \prod_{m \in \{C, P\}} \text{Poisson}(\lambda_{gm}).$$

The $M_1$ model is fitted for different values of $G$ between 1 and $G_{max}$ to achieve the best clustering model.

$M_2$ (Non-Clustering Model) which assumes that the proposed variable is not useful for clustering, but is potentially linked to the clustering variables through a Poisson GLM, that is,

$$X_n^C \sim \sum_{g=1}^{G} \tau_g \prod_{m \in C} \text{Poisson}(\lambda_{gm})$$
$$X_n^P | (X_n^C = x_n^C, Z_{ng} = 1) \sim \text{PoissonGLM}(x_n^{(C)}),$$

where Poisson GLM states that

$$\log \mathbb{E}[X_n^P | X_n^C = x_n^C, Z_{ng} = 1] = \alpha + \beta^\top x_n^C.$$

In order to avoid non significant terms in the Poisson GLM model, a standard stepwise variable selection approach (using BIC as the variable selection criterion) is considered. Thus, the proposed variable $X_n^P$ will be dependent on only a subset $X_n^R$ of the clustering variables $X_n^C$. We note that $G$ is fixed in the non-clustering model, because an optimal value for $G$ is previously chosen.

The clustering and non-clustering models are represented as graphical models in Figure 2.

Thus, there is two reasons for which $M_2$ can be preferred to $M_1$: either $X_n^P$ does not contain information about the latent clustering variable at all (ie. $X_n^R = \varnothing$), or $X_n^P$ does not add further useful information about the clustering given the information already contained in the current clustering variables. In the first situation, we say that $X_n^P$ is an irrelevant variable, because it contains no clustering information. In the second situation, we say that $X_n^P$ is a redundant variable because it contains no extra information about the clustering beyond the current clustering variables ($X_n^C$).

Additionally, both models assume the same form for the conditional distribution for $X_n^O | (X_n^C, X_n^P)$ and whose form doesn't need to be explicitly specified because it doesn't affect the model choice.

Variable $P$ is added to $C$ if the clustering model ($M_1$) is preferred to the non-clustering model ($M_2$). In order to compare $M_1$ and $M_2$, following (Dean and Raftery 2010), we consider the Bayes Factor:

$$B_{1,2} = \frac{p(X|M_1)}{p(X|M_2)}$$

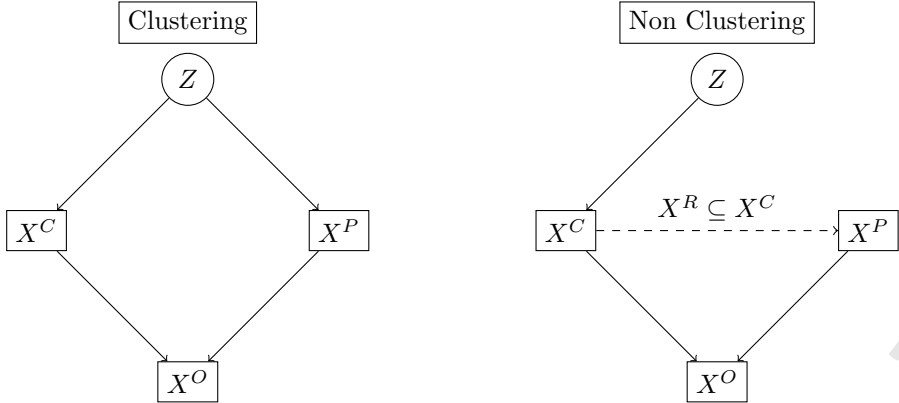

Figure 2: Graphical model representations of the clustering and non-clustering models.

¹²⁰ which is asymptotically approximated (Fop, Smart, and Murphy 2017; Kass and Raftery 1995) using
¹²¹ the difference of the BIC criteria for both models:

$$2 \log B_{1,2} \simeq BIC_{M_1} - BIC_{M_2}.$$

¹²² The same modelling framework can be used for removing variables from the current set of clustering
¹²³ variables.

## 4.2  Interpretation

¹²⁵ Comparing $M_1$ and $M_2$ is equivalent to comparing the following $X_n^P|(X_n^C = x_n^C)$ structures.

¹²⁶ The $M_1$ (Clustering Model) assumes that,

$$X_n^P|(X_n^C = x_n^C) \sim \sum_{g=1}^{G} \mathbb{P}\{Z_{ng} = 1|X_n^C = x_n^C\}\text{Poisson}(\lambda_{gm}),$$

¹²⁷ where

$$\mathbb{P}\{Z_{ng} = 1|X_n^C = x_n^C\} = \frac{\tau_g \prod_{m=1}^{M} \phi(x_{nm}, \lambda_{gm})}{\sum_{h=1}^{G} \tau_h \prod_{m=1}^{M} \phi(x_{nm}, \lambda_{hm})}.$$

¹²⁸ Whereas, the $M_2$ (Non-Clustering Model) assumes that,

$$X_n^P|(X_n^C = x_n^C) = \text{PoissonGLM}(x_n^C).$$

¹²⁹ The method contrasts which of conditional model structures is better describing the distribution of
¹³⁰ the proposed variable $X^P$. The clustering model ($M_1$) uses a mixture model, with covariate dependent
¹³¹ weights, for the conditional model whereas the non-clustering model ($M_2$) is a Poisson generalized
¹³² linear model. The model selection criterion chooses the model that best models this conditional
¹³³ distribution.

## 4.3  Stepwise selection algorithm

### 4.3.1  Screening variables: Initialization

¹³⁶ We start with an initial choice of $C$ by first screening each individual variable by fitting a mixture of
¹³⁷ univariate Poisson distributions (eg. Everitt and Hand 1981, chap. 4.3),

$$X_{nm} \sim \sum_{g=1}^{G} \tau_g \text{Poisson}(\lambda_{gm}), \text{ for } G = 1, 2, \dots, G_{max}.$$

The initial set of variables is set to be those variables where any model with $G > 1$ is preferred to the $G = 1$ model.

### 4.3.2 Stepwise algorithm: Updating

We consider a stepwise algorithm which alternates between adding and removing steps. In the removal step, all the variables in $X^C$ are examined in turn to be removed from the set. In the adding step, all the variables in $X^O$ are examined in turn to be added to the clustering set.

The algorithm also performs the selection of the number $G$ of clusters finding at each stage the optimal combination of clustering variables and number of clusters. The procedure stops when no change has been made to the set $X^C$ after consecutive exclusion and inclusion steps.

With the present stepwise selection algorithm, it can occur that during the process, we get back on a solution (a set of clustering variable) already explored. Since our algorithm is not stochastic, we fall into an infinite cycle. In this situation the algorithm is stopped, and the best solution according to BIC among the solution of the cycle is kept.

The following pseudo-code summarizes our stepwise algorithm:

ALGORITHM Stepwise
BEGIN
initialize $X^C$
WHILE $X^C$ changes:
- for all variable $X_j$ which are not in $X^C$
- estimate $M_1$ on $X^C \cup X_j$ and select the best $G$
- estimate $M_2$ with the model for $X^C$ (with G selected at the previous step) and a Poisson regression for $X_j$ given $X^C$
- add $X_j$ in $X^C$ if $BIC_{M_1} > BIC_{M_2}$
- for each $X_j$ in $X^C$
- estimate $M_2$ on $X^C \setminus X_j$, select the best $G$ and use a Poisson regression for $X_j$ given $X^C \setminus X_j$
- estimate $M_1$ on $X^C$ (with G selected at the previous step)
- remove $X_j$ from $X^C$ if $BIC_{M_2} > BIC_{M_1}$
- test for infinite loop
ENDWHILE
return $X^C$ and $M_1$ estimate
END

## 5 Simulation study

In this section, we evaluate the proposed variable selection method through three different simulation scenarios. We start with an illustrative example in which, using a data set simulated according to the proposed model, we show how to perform the variable selection.

Then, simulation studies are performed to evaluate the behavior of the proposed selection method, when the data are simulated according to the proposed model (Scenario1) and when the model assumptions are violated. In Scenario2, the link between $X^R$ and $X^C$ is no longer a Poisson GLM but a more complex model. In Scenario3, the clustering variables are no longer conditionally independent.

## 5.1 Illustrative example

In the first simulation setting we consider 10 Poisson random variables. Variables $X_1$, $X_2$, $X_3$ and $X_4$ are the clustering variables, distributed according to a mixture of $G = 3$ independent Poisson mixture distributions with mixing proportions 0.4, 0.3, 0.3. Variables $X_5$, $X_6$ and $X_7$ are redundant variables, each one generated dependent on the clustering variables. These three variables are linked to the four first ones through a Poisson GLM. The last three variables, $X_8$, $X_9$ and $X_{10}$ are irrelevant variables not related to the previous ones. Table 1 shows the parameter of the Poisson distribution for each variable and each cluster.

Table 1: True values of component parameters (Scenario 1)

|         | $\lambda_{g1}$ | $\lambda_{g2}$ | $\lambda_{g3}$ | $\lambda_{g4}$ | $\lambda_{g5}$ | $\lambda_{g6}$ | $\lambda_{g7}$ | $\lambda_{g8}$ | $\lambda_{g9}$ | $\lambda_{g10}$ |
|---------|------|------|------|------|------|------|------|------|------|------|
| $g = 1$ | 1 | 1 | 1 | 1 | $\lambda_{g5}$ | $\lambda_{g6}$ | $\lambda_{g7}$ | 4 | 2 | 1 |
| $g = 2$ | 2 | 2 | 1 | 4 | $\lambda_{g5}$ | $\lambda_{g6}$ | $\lambda_{g7}$ | 4 | 2 | 1 |
| $g = 3$ | 4 | 4 | 4 | 4 | $\lambda_{g5}$ | $\lambda_{g6}$ | $\lambda_{g7}$ | 4 | 2 | 1 |

with $\lambda_{g5} = \exp(0.2X_2)$, $\lambda_{g6} = \exp(0.2X_1 - 0.1X_2)$ and $\lambda_{g7} = \exp(0.1(X_1 + X_3 + X_4))$.

Below is the result obtained for one data set of size $N = 400$. The evaluation criteria is the selected features (true one are $X_1$ to $X_4$) and the Adjusted Rand Index (Rand 1971; Hubert and Arable 1985) obtained with the selected variables in comparison to those obtained with the full set of variables and with the true clustering variables.

The independent Poisson mixture model was fitted to the simulated data with $N = 400$ rows and $P = 10$ columns. Models with $G = 1$ to $G = 10$ were fitted using the EM algorithm.

The values of BIC for the independent Poisson mixture model are plotted in Figure 3.

The model with the highest BIC has $G = 3$ components and the resulting estimates of $\tau$ and $\lambda$ are given as:

Table 2: Estimates of the mixing proportions and component parameters.

|         | $\tau_g$ | $\lambda_{g1}$ | $\lambda_{g2}$ | $\lambda_{g3}$ | $\lambda_{g4}$ | $\lambda_{g5}$ | $\lambda_{g6}$ | $\lambda_{g7}$ | $\lambda_{g8}$ | $\lambda_{g9}$ | $\lambda_{g10}$ |
|---------|------|------|------|------|------|------|------|------|------|------|------|
| $g = 1$ | 0.29 | 4.09 | 4.00 | 4.15 | 4.34 | 2.51 | 1.87 | 3.95 | 4.04 | 1.85 | 1.12 |
| $g = 2$ | 0.42 | 2.04 | 2.11 | 1.34 | 3.74 | 1.64 | 1.27 | 2.00 | 3.91 | 2.06 | 0.96 |
| $g = 3$ | 0.29 | 0.93 | 0.88 | 1.08 | 0.96 | 1.13 | 1.01 | 1.16 | 3.82 | 2.02 | 1.00 |

A look at Table 1 of true values allows us to say that these estimates are correct (except for label switching).

Let start by initializing the stepwise algorithm.

```
fit_screen <- poissonmix_screen(x, G = 1:Gmax)
jchosen <- fit_screen$jchosen
```

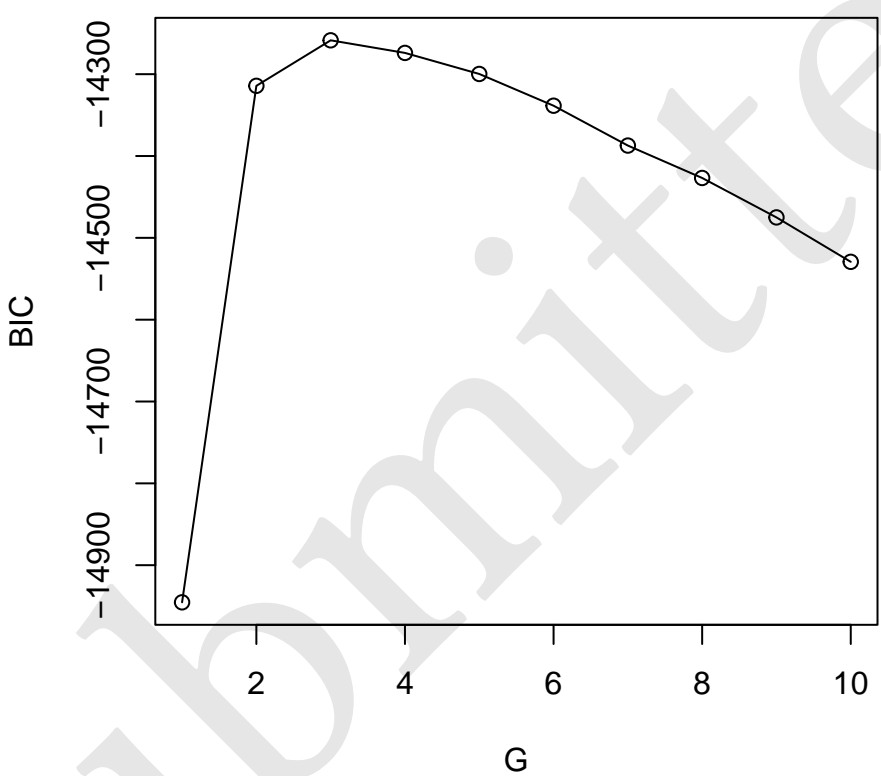

Figure 3: Bayesian Information Criterion (BIC) for the independent Poisson mixture model.

198 The variables selected by the screening procedure are {1, 2, 3, 4, 6, 7}.

199 Now, we execute the stepwise selection algorithm:

```
fit <- poissonmix_varsel(x, jchosen=jchosen, G = 1:Gmax)
```

200 [1] "Initial Selected Variables: 1,2,3,4,6,7"
201 [1] "Iteration: 1"
202 [1] "Add Variable: NONE 10 BIC Difference: -13.2"
203 [1] "Remove Variable: 6  BIC Difference: 83.7"
204 [1] "Current Selected Variables: 1,2,3,4,7"
205 [1] "Iteration: 2"
206 [1] "Add Variable: NONE 9 BIC Difference: -10.6"
207 [1] "Remove Variable: 7  BIC Difference: 50.1"
208 [1] "Current Selected Variables: 1,2,3,4"
209 [1] "Iteration: 3"
210 [1] "Add Variable: NONE 10 BIC Difference: -10.5"
211 [1] "Remove Variable: NONE 3 BIC Difference: -26.8"
212 [1] "Current Selected Variables: 1,2,3,4"

213 Note that the computing time is about 5 minutes on a laptop with 2.3 GHz Intel Core i7 processor
214 and 32Go of RAM.

215 The final chosen variables are {1, 2, 3, 4}.

216 Finally, the ARI obtained with the selected variables, which turn out to be the true clustering variable,
217 is 0.594 whereas it is 0.432 with all the variables.

## 5.2   Scenarios of simulation

219 In this section the three scenario of simulation are described. The first scenario is similar to the
220 previous illustrative example.

221 The second scenario is similar to the first one, except for variables $X_5$, $X_6$ and $X_7$ which are still
222 redundant but linked to the true clustering variables through linear, quadratic and exponential
223 term in an identity link function, respectively, and not a Poisson GLM with logarithm link function.
224 More precisely, $X_5$, $X_6$ and $X_7$ have Poisson distribution of respective parameter $\lambda_{g5} = \exp(2X_2)$,
225 $\lambda_{g6} = \exp(X_1^2 + X_3)$ and $\lambda_{g7} = \exp(\exp(0.1(X_1 + X_3 + X_4)))$. Thus, the data are simulated from a
226 model which does not satisfy assumptions of model $M_2$.

227 The third scenario is similar to the second one, but some dependence between the clustering variables
228 $X_1$ and $X_2$ is introduced, in order to create some redundancy among the true clustering variables.
229 For this, $X_1$ and $X_2$ are simulated as in the previous setting, and a same term is added to both of
230 these variables (simulated according a Poisson distribution of parameter 2) .

## 5.3   Results

232 Table 3 shows the number of times, among the 100 simulated data sets, that each variable is selected.
233 For Scenario 1, the model selection procedure perform perfectly, selecting each time only the true
234 clustering variables. For Scenario 2, due to the fact the link between the redundant and the true
235 clustering variables is not a standard Poisson GLM, the variable selection is perturbed and variables
236 $X_5$ is sometimes selected. For Scenario 3, the results is that the dependency between $X_1$ and $X_2$
237 perturb the variable selection, and only one of them is selected (and even sometimes none of them).
238 Redundant variables $X_5$ and $X_6$, which are linked to the clustering variables but with a linear link,
239 are also sometimes selected.

Table 3: Number of selection for each variable, simulation setting number 3.

|  | $X_1$ | $X_2$ | $X_3$ | $X_4$ | $X_5$ | $X_6$ | $X_7$ | $X_8$ | $X_9$ | $X_{10}$ |
|---|---|---|---|---|---|---|---|---|---|---|
| Scenario 1 | 100 | 100 | 100 | 100 | 0 | 0 | 0 | 0 | 0 | 0 |
| Scenario 2 | 97 | 100 | 90 | 98 | 44 | 0 | 0 | 0 | 0 | 0 |
| Scenario 3 | 48 | 35 | 89 | 88 | 65 | 34 | 3 | 0 | 0 | 0 |

Figure 4 plots the distribution of the ARI differences between the model with either the selected variables or all the variables, and the one with the true clustering variables. These plots shows that for all scenarios, the ARI of the model with the selected variables (left boxplot of each plot) are always closest to the optimal ARI (obtained with the true clustering variables).

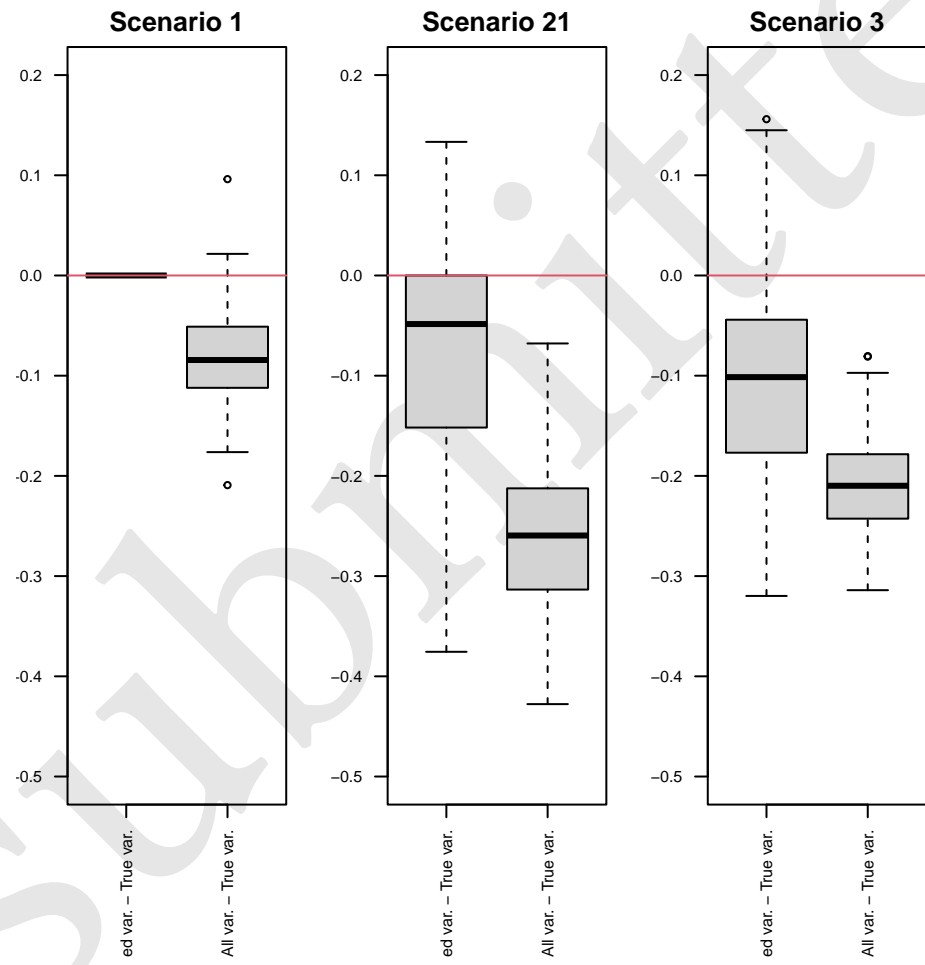

Figure 4: Distribution of the ARI differences with the model with the true clustering variables, for the model with the selected variables and the model with all variables.

Finally Figure 5 plots the histogram of the difference of ARI with the selected variables and with all the variables. This plot illustrates the interest of variable selection on the clustering results, and indeed, for all the scenarios, the ARI is better with the selected variables than when using all the variables.

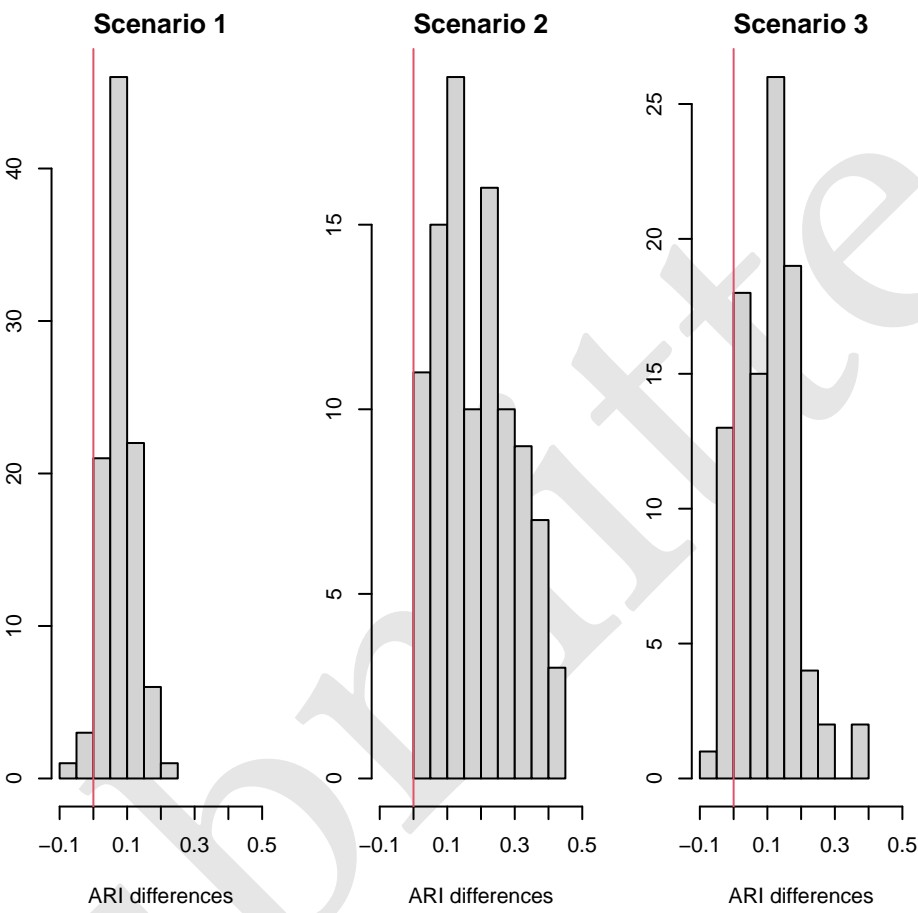

Figure 5: Distribution of the ARI differences for the model with the selected variables and the model with all variables.

## 6 International Ultrarunning Association Data

We apply the proposed procedure to the data from the 2012 International Ultrarunning Association World 24H Championships.

We start by initializing the stepwise algorithm, and find the variables selected by the screening procedure:

```
fit_screen <- poissonmix_screen(x, G = 1:Gmax)
jchosen <- fit_screen$jchosen
jchosen
```

```
[1]  3  5  6  7  8  9 11 12 13 14 15 16 17 18 19 20 21 22 23 24
```

We then execute the proposed stepwise selection algorithm (the computing time is about 26 minutes on a laptop with 2.3 GHz Intel Core i7 processor and 32Go of RAM):

```
fit <- poissonmix_varsel(x, jchosen = jchosen, G = 1:Gmax)
```

The final chosen variables found by the algorithm are:

```
[1]  9 10 11 12 14 15 16 17 18 19 20 21 22 24
```

The optimal number of clusters 6 has been chosen inside the stepwise selection algorithm. The same choice is obtained when looking for the best $G$ with the conditionally independent Poisson mixture on the selected variables (Figure 6).

In order to illustrate the results, we plot the cluster means according to the 24 variable mean parameters per cluster. For each variable not in the chosen variable set, a Poisson regression model is fitted with the chosen variables as predictors. Forward and backwards variable selection is conducted on this regression, if the regression model has any predictor variables, then the variable is called "redundant" and if the regression model has no predictor variables, then the variable is called "irrelevant". Figure 7 shows the cluster mean for each variable, where the label indicates if the variable is irrelevant for clustering ("I"), redundant ("R") or useful (the label is then the cluster number).

The variables discriminate the clusters pacing strategies of the runners are the number of laps covered during the last two thirds of the race (except during the 13th and 23rd hours). The number of laps covered during the first eight hours does not provide any additional clustering information, and even no information at all for the number of laps covered during the first hour.

Figure 8 plots the density map per clusters. Area of high density (red) indicates the hours and the corresponding average number of laps specific of each cluster.

Cluster 5 are clearly the most efficient runners. Looking at the running strategy in Figure 7 and Figure 8, we can see that they start as runners of Cluster 1 and Cluster 2, but they managed to keep a constant pace on the second part of the race, unlike those of the other two clusters which faltered. Runners of Cluster 3 has covered the fewest number of laps. Indeed, looking at their running strategy, we can see that most of these runners stop after the first third of the race. Cluster 6 is relatively similar to Cluster 3, but runners manage to continue running until half of the race is completed. Finally, Cluster 4 obtains slightly better results than Cluster 6, starting more carefully, and managing to run until the end of the race, even if the pace of the last hours is not very constant.

Finally, Figure 9 shows boxplots of the total number of loops covered by the runners in each of the clusters.

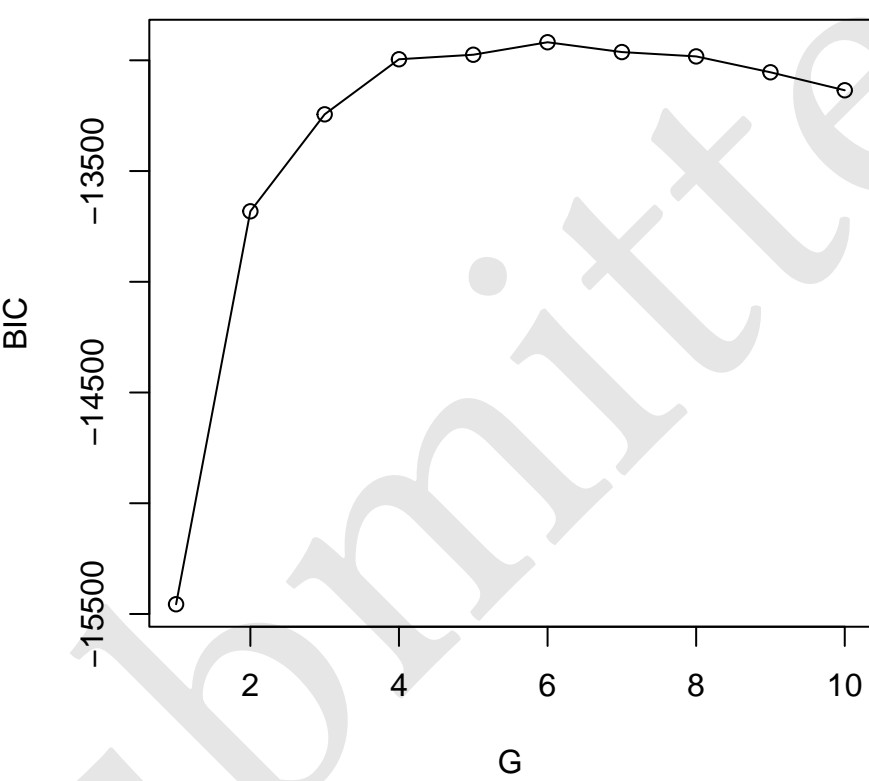

Figure 6: Bayesian Information Criterion (BIC) for the independent Poisson mixture model with the seleceted variables.

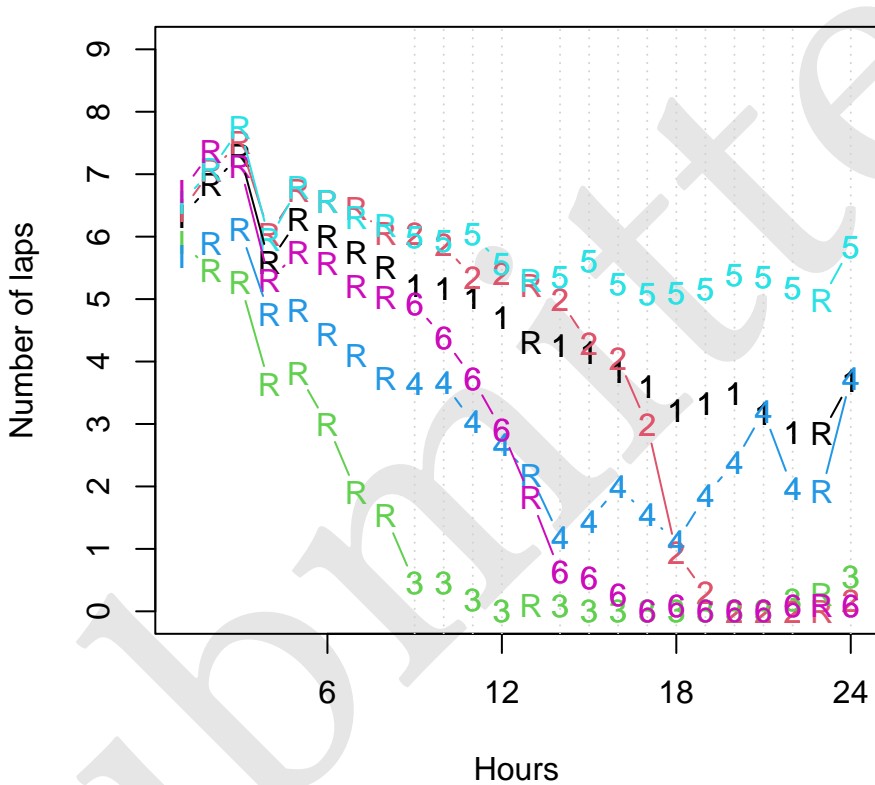

Figure 7: Cluster means and usefulness of the variables.

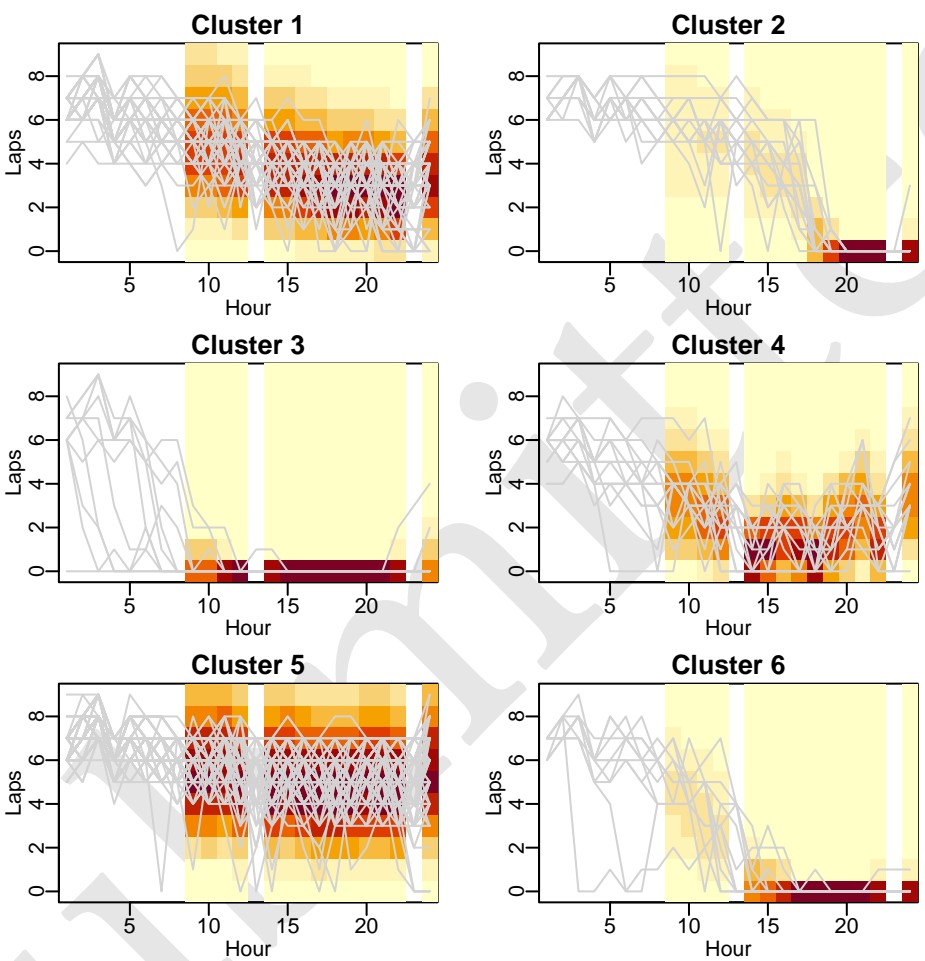

Figure 8: Density maps per cluster

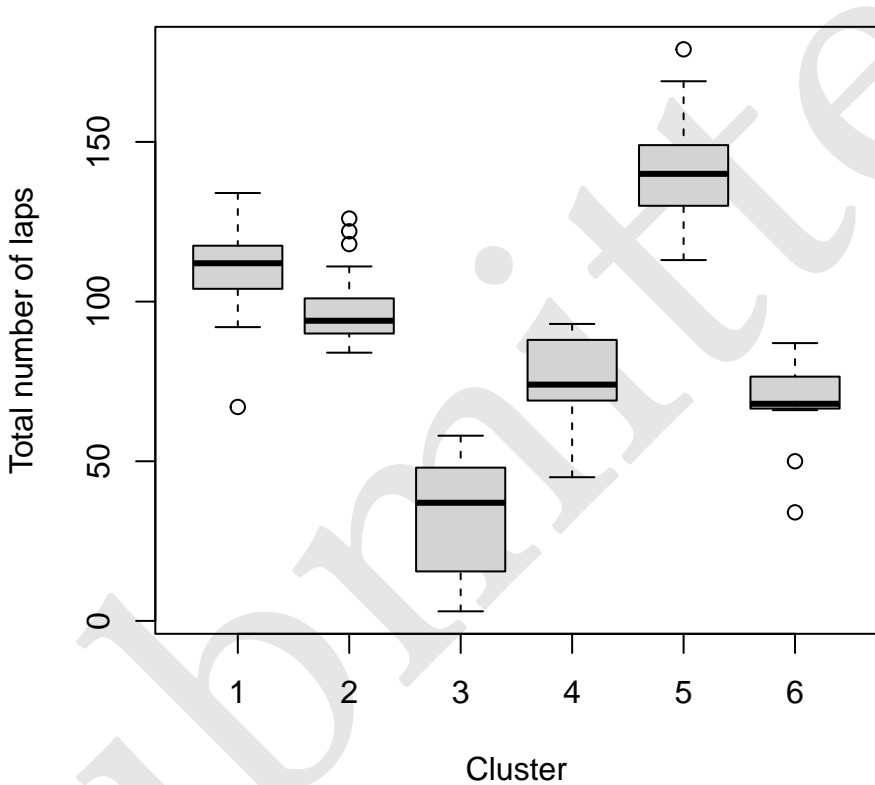

Figure 9: Number of loops covered by the runners of each clusters.

## 7  Discussion

A method for clustering and variable selection for multivariate count data has been proposed. The method is shown to give excellent performance on both simulated and real data examples. The method selects set of relevant variables for clustering and other variables are not selected if they are irrelevant or redundate for clustering purposes.

The proposed method is shown to give interesting insights in the application domain, where some clusters members are shown to perform better overall to others and the benefits of constant (or near constant pacing) are shown.

The level of variable selection is determined by the relative performance of the two models (as shown in Section 4.2) is compared. Alternative models to the Poisson GLM model which have greater flexibility could lead to a smaller set of selected variables than the proposed method achieves. This is a topic for future research.

The proposed method is based on a conditionally independent Poisson mixture model for the selected variables. It could be argued that the conditional independence assumption is unrealistic in the application. Hand and Yu (2001) consider the implication of incorrectly assuming conditional independence in a classification setting and show that it can make the group membership probabilities over confident. Furthermore, in the conditional independent Poisson mixture model, the number of clusters can be upwardly biased, where extra clusters are included to model dependence in the data. The approach taken in the paper could be extended to use other multivariate count distributions, including multivariate distributions without the conditional independence assumption (eg. Karlis 2018; Karlis and Meligkotsidou 2007; Inouye et al. 2017b).

The code for the proposed approach is available as an R package at https://github.com/JuJacques/MultivariateCountData.

## 8  Acknowlegements

This work was supported by the Science Foundation Ireland Insight Research Centre (SFI/12/RC/2289_P2) and a visit to the Collegium – Institut d'Études Avancées de Lyon.

Agresti, Alan. 2013. *Categorical Data Analysis.* Third. Wiley Series in Probability and Statistics. Wiley-Interscience [John Wiley & Sons], Hoboken, NJ.

Bouveyron, Charles, Gilles Celeux, T. Brendan Murphy, and Adrian E. Raftery. 2019. *Model-Based Clustering and Classification for Data Science.* Cambridge Series in Statistical and Probabilistic Mathematics. Cambridge University Press, Cambridge. https://doi.org/10.1017/9781108644181.

Chiquet, Julien, Mahendra Mariadassou, and Stéphane Robin. 2021. "The Poisson-Lognormal Model as a Versatile Framework for the Joint Analysis of Species Abundances." *Frontiers in Ecology and Evolution* 9: 188. https://doi.org/10.3389/fevo.2021.588292.

Dean, Nema, and Adrian E. Raftery. 2010. "Latent Class Analysis Variable Selection." *Annals of the Institute of Statistical Mathematics* 62 (1): 11–35. https://doi.org/10.1007/s10463-009-0258-9.

Dempster, Arthur P., Nan M. Laird, and Donald B. Rubin. 1977. "Maximum Likelihood from Incomplete Data via the EM Algorithm." *Journal of the Royal Statistical Society: Series B* 39: 1–38. https://doi.org/10.1111/j.2517-6161.1977.tb01600.x.

Everitt, Brian S., and David J. Hand. 1981. *Finite Mixture Distributions.* Chapman & Hall.

Fop, Michael, and Thomas Brendan Murphy. 2018. "Variable Selection Methods for Model-Based Clustering." *Statistics Surveys* 12: 18–65. https://doi.org/10.1214/18-SS119.

Fop, Michael, Keith Smart, and Thomas Brendan Murphy. 2017. "Variable Selection for Latent Class Analysis with Application to Low Back Pain Diagnosis." *Annals of Applied Statistics.* 11: 2085–115.

Frühwirth-Schnatter, Sylvia, Gilles Celeux, and Christian P. Robert. 2018. *Handbook of Mixture Analysis*. Chapman; Hall/CRC. https://doi.org/10.1201/9780429055911.

Hand, David J., and Keming Yu. 2001. "Idiot's Bayes—Not so Stupid After All?" *International Statistical Review* 69 (3): 385–98. https://doi.org/https://doi.org/10.1111/j.1751-5823.2001.tb00465.x.

Hartigan, John A., and M. Anthony Wong. 1979. "A k-Means Clustering Algorithm." *Applied Statistics* 28 (1): 100–108.

Hubert, Lawrence, and Phipps Arable. 1985. "Comparing Partitions." *Journal of Classification* 2 (1): 193–218. https://doi.org/10.1007/BF01908075.

Inouye, David I., Eunho Yang, Genevera I. Allen, and Pradeep Ravikumar. 2017a. "A Review of Multivariate Distributions for Count Data Derived from the Poisson Distribution." *WIREs Computational Statistics* 9 (3): e1398. https://doi.org/https://doi.org/10.1002/wics.1398.

———. 2017b. "A Review of Multivariate Distributions for Count Data Derived from the Poisson Distribution." *WIREs Computational Statistics* 9 (3): e1398. https://doi.org/10.1002/wics.1398.

Karlis, Dimitris. 2018. "Mixture Modelling of Discrete Data." In *Handbook of Mixture Analysis*, edited by Sylvia Frühwirth-Schnatter, Gilles Celeux, and Christian P. Robert, 193–218. CRC Press.

Karlis, Dimitris, and Loukia Meligkotsidou. 2007. "Finite Mixtures of Multivariate Poisson Distributions with Application." *Journal of Statistical Planning and Inference* 137 (6): 1942–60. https://doi.org/10.1016/j.jspi.2006.07.001.

Kass, Robert E., and Adrian E. Raftery. 1995. "Bayes Factors." *Journal of the American Statististical Association* 90 (430): 773–95. https://doi.org/10.1080/01621459.1995.10476572.

Maugis, Cathy, Gilles Celeux, and Marie-Laure Martin-Magniette. 2009. "Variable Selection in Model-Based Clustering: A General Variable Role Modeling." *Computational Statistics & Data Analysis* 53 (11): 3872–82. https://doi.org/10.1016/j.csda.2009.04.013.

McLachlan, Geoffrey, and David Peel. 2000. *Finite Mixture Models*. New York: Wiley. https://doi.org/10.1002/0471721182.

McParland, Damien, and Thomas Brendan Murphy. 2018. "Mixture Modelling of High-Dimensional Data." In *Handbook of Mixture Analysis*, edited by Sylvia Frühwirth-Schnatter, Gilles Celeux, and Christian P. Robert, 247–80. CRC Press.

Raftery, Adrian E., and Nema Dean. 2006. "Variable Selection for Model-Based Clustering." *Journal of the American Statistical Association* 101 (473): 168–78. https://doi.org/10.1198/016214506000000113.

Rand, William M. 1971. "Objective Criteria for the Evaluation of Clustering Methods." *Journal of the American Statististical Association* 66 (336): 846–50.

Rau, Andrea, Cathy Maugis-Rabusseau, Marie-Laure Martin-Magniette, and Gilles Celeux. 2015. "Co-expression analysis of high-throughput transcriptome sequencing data with Poisson mixture models." *Bioinformatics* 31 (9): 1420–27. https://doi.org/10.1093/bioinformatics/btu845.

Schwarz, Gideon. 1978. "Estimating the Dimension of a Model." *The Annals of Statistics* 6 (2): 461–64. https://doi.org/10.1214/aos/1176344136.

Silva, Anjali, Steven J. Rothstein, Paul D. McNicholas, and Sanjeena Subedi. 2019. "A Multivariate Poisson-Log Normal Mixture Model for Clustering Transcriptome Sequencing Data." *BMC Bioinformatics* 20 (1): 394. https://doi.org/10.1186/s12859-019-2916-0.

White, Arthur, and Thomas Brendan Murphy. 2016. "Exponential Family Mixed Membership Models for Soft Clustering of Multivariate Data." *Advances in Data Analysis and Classification* 10: 521–40.

