# OpenReview forum: "Model-Based Clustering and Variable Selection for Multivariate Count Data"
_Computo — Accepted by Computo_

### Review · Reviewer_v2H5 · 2025-01-24

**Summary Of Contributions:**

The authors propose a novel model-based clustering method for count data based on a mixture of Poisson distributions. Moreover, they present a model selection algorithm to identify the variables informative for the resulting clustering. Through simulation studies and one real-world application, the author show the use and performance of their method, implemented in R.

**Audience:**

Yes

**Broader Impact Concerns:**

None.

**Claims And Evidence:**

Yes

**Requested Changes:**

In my opinion, the authors should address the following major points.

1. Can you comment on the independence assumption in the proposed Independent Poisson Mixture? In particular, it seems that in the context of the application two consecutive hours could be dependent on each other. Simulation setting 3 partially addresses this point, but clearly shows that the proposed variable selection step is negatively affected. While the authors show that the ARI of the method is still reasonably good, they should at least discuss what the independence assumption means in the context of their application and clearly highlight this limitation in the discussion.
2. From Section 4.1 and Figure 2, it looks like an implicit assumption of the approach is that X^O is conditionally independent of Z given {X^C, X^P}, i.e., that X^O influences the clustering only through X^C or X^P. The authors should make this assumption explicit. Moreover, can the authors comment on the practical implications of this in the context of their stepwise variable selection algorithm? While it looks reasonable to me in a backward selection procedure, I fail to understand how to justify this in a forward procedure, in which some components of X^O will become X^P in a next step.
3. How do the authors choose the variable entry order in the stepwise selection procedure? Does the order influence the final results? Perhaps a simulation comparing the results of different orderings could shed some light on this.
4. Reproducibility: while the paper is reproducible and the method is implemented in R functions linked to the paper, these functions lack documentation. I suggest that the authors implement their functions in an installable R package, to be made publicly available on github (at minimum) or in proper repositories such as CRAN (preferred).

Minor issues:

- There are several typos and grammatical errors, e.g.
     - Page 2, line 21: genomic -> genomics
     - Page 4, line 70: consists starting -> starts
     - Page 4, lines 82-83: use \citet instead of \citep
     - Page 5, line 101: there is a missing verb after “avoid to”
     - Page 6, line 133: “the any” -> “any”
     - Page 8, line 190: 4 -> 100 \
     I suggest that the authors carefully proof-read the paper
- Can the authors briefly describe the parameter estimation of model M1 of Section 4.2?
- What are the true cluster means in the simulations? Please add this in the scenario descriptions.
- Please, add the middle label in the x axis of the boxplot of Figure 4.
- Why is there a star in M2 at page 12, line 206?

**Strengths And Weaknesses:**

The main strength of the proposed approach is that it respects the count nature of the data, which is important in many applications, especially when counts are low and the naive approach of using Gaussian-based methods may fail.

The main weakness is in my opinion represented by the independence assumption in the multivariate count distribution. Especially in the context of the presented application, it seems that the number of laps completed at a given hour for each runner may be dependent at least on the neighboring hours, e.g., because of some latent runner characteristics or because the runner is getting tired or warming up.

---

> ### Author Response · Authors · 2025-03-21
> **Point by point responses**
>
> Requested Changes:
>
> In my opinion, the authors should address the following major points.
>
> 	1	Can you comment on the independence assumption in the proposed Independent Poisson Mixture? In particular, it seems that in the context of the application two consecutive hours could be dependent on each other. Simulation setting 3 partially addresses this point, but clearly shows that the proposed variable selection step is negatively affected. While the authors show that the ARI of the method is still reasonably good, they should at least discuss what the independence assumption means in the context of their application and clearly highlight this limitation in the discussion.
>
> We discuss in the conclusion, but we have added extra some discussion.
>
> 	2      From Section 4.1 and Figure 2, it looks like an implicit assumption of the approach is that X^O is conditionally independent of Z given {X^C, X^P}, i.e., that X^O influences the clustering only through X^C or X^P. The authors should make this assumption explicit. Moreover, can the authors comment on the practical implications of this in the context of their stepwise variable selection algorithm? While it looks reasonable to me in a backward selection procedure, I fail to understand how to justify this in a forward procedure, in which some components of X^O will become X^P in a next step.
>
> This is needed so that every iteration we are modeling the full data. BIC can only be used for model comparison if every time it is used it has the same data. We assume that X^O is conditionally independent of Z given X^C and X^P. We also assume the same distribution for p(X^O | X^C, X^P) in both models too, which means that this term cancels in any BIC comparison. Because of the iterative nature of the algorithm the decision on whether to add/remove a variable is a local decision based on the current chosen and proposed variables. In the final model we only explicitly model the chosen variables using a conditionally independent Poisson mixture and the other variables have no role.
>
> 	3	How do the authors choose the variable entry order in the stepwise selection procedure? Does the order influence the final results? Perhaps a simulation comparing the results of different orderings could shed some light on this.
>
> The order has no impact : Each variable is tested in turn for addition (removal) to the model, and the one that best improves the BIC is added.
>
> 	4	Reproducibility: while the paper is reproducible and the method is implemented in R functions linked to the paper, these functions lack documentation. I suggest that the authors implement their functions in an installable R package, to be made publicly available on github (at minimum) or in proper repositories such as CRAN (preferred).
>
> The GitHub repository is now public, and contains an R package.
>
> Minor issues:
>
> 	•	There are several typos and grammatical errors, e.g.
> 	◦	Page 2, line 21: genomic -> genomics
> 	◦	Page 4, line 70: consists starting -> starts
> 	◦	Page 4, lines 82-83: use \citet instead of \citep
> 	◦	Page 5, line 101: there is a missing verb after “avoid to”
> 	◦	Page 6, line 133: “the any” -> “any”
> 	◦	Page 8, line 190: 4 -> 100  I suggest that the authors carefully proof-read the paper
>
> Fixed
>
> 	•	Can the authors briefly describe the parameter estimation of model M1 of Section 4.2?
>
> Fixed
>
> 	•	What are the true cluster means in the simulations? Please add this in the scenario descriptions.
>
> Fixed
>
> 	•	Please, add the middle label in the x axis of the boxplot of Figure 4.
>
> Fixed
>
> 	•	Why is there a star in M2 at page 12, line 206?
>
> This was from previous notation and it has been fixed.

---

### Review · Reviewer_FafF · 2025-01-28

**Summary Of Contributions:**

The submission introduces a model for clustering and variable selection of multivariate count data. The variable selection step is intended to both improve clustering and classify the variables into *relevant*, *redundant* and *irrelevant* for the clustering.

The *clustering part* consists of a mixture of poisson random variables. The Poisson variables are modeled as independent conditional on the cluster and the model parameters are estimated using the E-M algorithm gives simple and natural update rules in this setting. The (unknown) number of clusters is selected using the BIC criteria.

The *variable selection* uses a stepwise model selection approach where a set of *active* variables is iteratively updated by adding the *best* non active variable / removing the worst active variable as long as addition / removal improves the BIC (BIC changes being seen as a aproximation of Bayes factors) until either (i) the active set doesn't change any more or it gets stuck in a cycle (*e.g.* adding $X_1$, removing $X_2$, removing $X_1$, adding $X_2$, etc), in which case the model with the best BIC in the cycle is selected.

The authors then run a simulation study showing that the inference reaches very good performance when the model is well-specified and is robust to some amount of model-misspecification (dependence between the relevant variables, model misspecification for the link between relevant and redundant variables).

They finally apply it to their motivating example of ultrarunning, where the count variables is the 24-dimensional vector of the number of loops completed by the runners for each hour of the 24 hours-long race. The method identifies 6 clusters with constrated and interpretable running patterns.

**Audience:**

Yes

**Broader Impact Concerns:**

No broader impact concerns

**Claims And Evidence:**

Yes

**Requested Changes:**

**Major comments:**

- Please specify whether a unified statistical model is used for clustering and variable selection (if so change the text accordingly) or two different statistical models are considered.
- In the variable selection part, please clarify how variable screening for addition to / removal from the active set is combined with the choice of the number of clusters.
- In the simulation study, combine the results from all simulation settings into a single table and a single figure to make it easy to compare results across settings.
- In the simulation study, give more details about the parameter values used for the three different settings.
- Please add details about computing time and code availability

**Suggestions for improvements:**

- In the simulation study, consider using violin plots of $\Delta$ARI (like those shown in the histograms) to emphasize the cost of not knowing the true clustering variables (true versus all and true versus selected). Histograms of $\Delta$ARI could then be included in this figure (selected versus all) to further emphasize the benefits of variable selection. This however comes at the lost of showing the actual ARI values.
- In the application, please consider adding the BIC graph to show how much better 6 fares than other numbers of clusters.
- In the application, please consider using small multiple plots to show the individual distributions of $X_1, \dots, X_{24}$ in each group, using for example a bubble chart to deal with the discrete nature of the count. This would help assess if the data are well modeled using independent Poisson distributions in each cluster.

**Minor comments:**

- l.213: true clsutering $\rightarrow$ true clustering
- Caption of Figure 7: simulation setting number S $\rightarrow$ simulation setting number 2

**Strengths And Weaknesses:**

**Strengths**

Each step (clustering and variable selection) is presented clearly at a high level. The simulation study demonstrates good performances of the method and addresses the important problem of model misspecification and robustness of the results. The motivating example makes for a very relevant application of the method which lends itself nicely to results discussion and interpretation.

**Weaknesses**

The split between clustering and variable selection makes it hard to understand the overall model used (if any). In the clustering part, the $M$ variables $X^M$ are assumed to be independent conditionally on the cluster. By contrast, in the variable selection part (and figure 2), the graphical model shows that the $M$ variables are partitioned into clustering variables ($X^C$), proposed variables $X^P$ that are either redundant or irrelevant if they depend (or not) on $X^C$ through a Poisson-GLM and other variables ($X^O$) whose distribution depends on both $X^C$ and $X^P$. This second model is at odds with the first one and it would be nice to state whether both parts take place in the same theoretical framework (and present it before declining it for the clustering part and the variable selection part) or if both are disconnected.

l.111-112: *The conditional distribution for $X^O | X^C, X^P$ whose form doesn't need to be specified*. Yet the decision rule for including a variable into the active set relies on the Bayes Factor which is approximated by the difference in BIC values and thus assumes a model for $X^O | X^C, X^P$ (unless $X_O$ is completely ignored when computing the BIC values).

l.139-141: This paragraph is a bit vague and lends itself to too many intepretations. How exactly is the optimal combination for clustering variables and number of clusters chosen. Are all combinations ($G_\max \times (M - |C|)$ for the inclusion step and $G_\max \times |C|$ for the removal step) evaluated systemically ? Or is the active set updated (for the current number cluster) before updating the number of clusters (for the current active set). An algorithm in pseudo-code would be very helpful here.

Section 5: Details on the simulations are a bit sparse (although the code is available and can be examined in the qmd file). It would be helpful to describe how the three clusters differ in terms of parameters. Likewise for the different settings of model misspecification, giving more details (for example the level of dependence in simulation setting 4) would help the reader. Finally Tables 2 to 4 could be merged (and likewise for figures 4, 6 and 8 one the one hand and figures 5, 7 and 9 on the other hand) to ease comparisons between settings. The boxplots could also directly show difference in ARI between the oracle (true variables) and the two variants (with and without variable selection) to emphasize the benefits of variable selection and get rid of the histograms (Figures 5, 7 and 9).

Details on computing time and code availability are missing. It would be very helpful to know if the methods runs in seconds or in minutes on the motivating dataset.

---

> ### Author Response · Authors · 2025-03-21
> **Point by point response**
>
> Requested Changes:
>
> Major comments:
>
> 	•	Please specify whether a unified statistical model is used for clustering and variable selection (if so change the text accordingly) or two different statistical models are considered.
>
> The main goal of the paper is to do clustering of multivariate count data. In that sense, only one model is considered, the Conditionally Independent Poisson Mixture described in Section 3.
> Nevertheless, since all the available variables are not necessary useful to cluster the data, Section 4 proposes a procedure to select the variables useful for clustering.
> Once these useful variables have been identified, the considered model is the Independent Poisson Mixture.
> We have added extra explanations at the end of the introduction.
>
> 	•	In the variable selection part, please clarify how variable screening for addition to / removal from the active set is combined with the choice of the number of clusters.
>
> On any iteration, the M_2 model is the same model as was chosen on the previous iteration + a GLM for the conditional distribution of x^P given x^C. So, the number of groups doesn’t need to be estimated in the M_2 model.
> For the M_1 model we do search over the number of clusters to get the best value of G. There is no need to reestimate G at the end of the procedure because this model is exactly equivalent to fitting the conditional independent Poisson mixture model to the selected variables only.
> So, in principle the number of clusters can change from one iteration to the next. It is not fixed. This is justified by us trying to find the “best” model on each iteration.
> We have added extra sentence about that after the definition of M_1 and M_2.
>
> 	•	In the simulation study, combine the results from all simulation settings into a single table and a single figure to make it easy to compare results across settings.
>
> Fixed
>
> 	•	In the simulation study, give more details about the parameter values used for the three different settings.
>
> Fixed
>
> 	•	Please add details about computing time and code availability
>
> The Github repository which contains the Computo submission is now public. We also built a R package which is available in the Github repository.
> We give the computing time for the simulation study (Scenario 1) and for the real data analysis.
>
> Suggestions for improvements:
>
> 	•	In the simulation study, consider using violin plots of ARI (like those shown in the histograms) to emphasize the cost of not knowing the true clustering variables (true versus all and true versus selected). Histograms of ARI could then be included in this figure (selected versus all) to further emphasize the benefits of variable selection. This however comes at the lost of showing the actual ARI values.
>
> We add a new plot for the ARI difference between the model with the selected/all variables and the model with the true variables.
>
> 	•	In the application, please consider adding the BIC graph to show how much better 6 fares than other numbers of clusters.
>
> The number of clusters has been chosen inside the stepwise algorithm. We reload this choice on the conditional independent Poisson Mixture on the set of selected variables and plot the corresponding values of BIC.
>
> 	•	In the application, please consider using small multiple plots to show the individual distributions of in each group, using for example a bubble chart to deal with the discrete nature of the count. This would help assess if the data are well modeled using independent Poisson distributions in each cluster.
>
> Thank you for this good idea. We added this new plot (Figure 8).
>
> Minor comments:
>
> 	•	l.213: true clsutering true clustering
>
> Fixed
>
> 	•	Caption of Figure 7: simulation setting number S simulation setting number 2
>
> Fixed

---

> > ### Comment · Reviewer_FafF · 2025-03-25
> > **Request for clarification of some points**
> >
> > I thank the authors for addressing most of my comment. I just have a few questions left as I'm still confused on the selection of $G$ and the variable set. You mention
> >
> > > On any iteration, the M_2 model is the same model as was chosen on the previous iteration + a GLM for the conditional distribution of x^P given x^C. So, the number of groups doesn’t need to be estimated in the M_2 model. For the M_1 model we do search over the number of clusters to get the best value of G. There is no need to reestimate G at the end of the procedure because this model is exactly equivalent to fitting the conditional independent Poisson mixture model to the selected variables only. So, in principle the number of clusters can change from one iteration to the next. It is not fixed. This is justified by us trying to find the “best” model on each iteration. We have added extra sentence about that after the definition of M_1 and M_2.
> >
> > Do I understand correctly that the workflow proceeds as:
> > 1. Fit several CIPM (one per value of $G$) to $(X^C, X^P)$ (M1) and select $\hat{G}_{C, P}$ using the BIC criteria
> > 2. For this value $\hat{G}_{C,P}$, fit (M2) to $(X^C, X^P)$
> > 3. Compute the Bayes factor $B_{1,2}^P$
> > 4. Among all variables $X^P$ for which $B_{1,2}^P > 1$, add the one corresponding to the highest $B_{1,2}^P$
> >
> > Or is the value $G$ considered in the second step $\hat{G}_{C}$ the one found in the previous iterations (when fitting M1 to $X^C$, before considering new candidate variables) ?  In any case, a small pseudo-code description of the update procedure in section 4.3.2 would help clarify the procedure.
> >
> > > L. 288 The code for the proposed approach will be made available as an R package.
> > Since the package is now available, you can update that sentence and point to the github repo

---

> > > ### Author Response · Authors · 2025-03-27
> > > **Answer**
> > >
> > > You correctly understand the workflow.
> > > As you suggested, we add a pseudo-code in Section 4.3.2.
> > > We thank you for this advice which make the paper clearer.
> > >
> > > We also give the link for the Github Repository.

---

### Comment · Action_Editor_aCR2 · 2025-01-28
**Rebuttal period**

Dear authors,

We have received two reports for your submission to Computo entitled “Model-Based Clustering and Variable Selection for Multivariate Count Data”.

A period of 6 weeks is allowed for discussion with the referees before they issue a final opinion. During this period, you can make any changes to your submission that you feel are necessary and that you are able to make. At the end of this period, a decision will be made, ranging from final acceptance to more substantial requests for modification.

Best regards

---

### Comment · Action_Editor_aCR2 · 2025-03-12
**Rebuttal period still in progres: please wait before submitting your official recommendation**

Dear reviewers,

You have received a notification about submitting your recommendation for this paper by April 4.

Please ignore this notification, as the authors have asked for two more weeks to complete their revision. We will ask you to submit your recommendation after this date.

---

### Comment · Action_Editor_aCR2 · 2025-04-18
**Please submit your recommendation**

At this stage of the process, you are asked to submit a "Decision Recommendation" for this paper, in the light of the response of the authors to the reviews.

Do not hesitate to reach out to me for any question.

---

### Comment · Action_Editor_aCR2 · 2025-05-28
**Final acceptance**

Thank you for this final revision. The paper is now formally accepted and we are ready move to the production step.

---

### Note · Reviewer_FafF · 2025-03-27

**Comment:**

The paper is clearly written, reproducible, falls in the scope of Computo (first category: "New methods with original stats/ML developments, or numerical studies that illustrate theoretical results in stats/ML") and would be of interest to readers working on mixture models for count data.

**Audience:**

Yes

**Claims And Evidence:**

Yes

**Decision Recommendation:**

Accept

---

### Note · Reviewer_v2H5 · 2025-05-06

**Comment:**

I thank the authors for responding to my comments and I find the revised version of the paper improved compared to the first submission. However, some of my previous comments were not adequately addressed, and I suggest that the authors address them in full before final publication.

In particular:

1. While the response of my previous "Point 2" is satisfactory, I would like to see a version of their response in the Discussion of the paper, since readers might have the same question about the role of $X^O$ in their algorithm.

2. The code in the Github repository is not a valid installable R package. I would suggest to create a proper installable package. Alternatively, the authors should refer to the code as "R code" and not as an "R package".

3. There are still several typos and grammatical issues in the paper for instance:
- Lines 32-33: this sentence needs rewriting
- Line 72: there are two verbs
- Line 84-85: \citet should be used instead of \citep
I believe that I already made these remarks in my first review and I suggest that the authors carefully proof read their manuscript and fix all the typos.

**Audience:**

Yes

**Claims And Evidence:**

Yes

**Decision Recommendation:**

Leaning Accept

---

### Decision · Action_Editor_aCR2 · 2025-05-07

**Recommendation:** Accept with minor revision

**Comment:**

Recommendation of Reviewer v2H5

> I thank the authors for responding to my comments and I find the revised version of the paper improved compared to the first submission. However, some of my previous comments were not adequately addressed, and I suggest that the authors address them in full before final publication.
>
> In particular:
>
> 1. While the response of my previous "Point 2" is satisfactory, I would like to see a version of their response in the Discussion of the paper, since readers might have the same question about the role of $X^O$ in their algorithm.
>
> 2. The code in the Github repository is not a valid installable R package. I would suggest to create a proper installable package. Alternatively, the authors should refer to the code as "R code" and not as an "R package".
>
> 3. There are still several typos and grammatical issues in the paper for instance:
> -  Lines 32-33: this sentence needs rewriting
> -  Line 72: there are two verbs
> -  Line 84-85: \citet should be used instead of \citep I believe that I already made these remarks in my first review and I suggest that the authors carefully proof read their manuscript and fix all the typos.

**Audience:**

This paper clearly fits the scope of Computo very well. It will be of interest to readers working on mixture models for count data.

**Claims And Evidence:**

Dear authors,

Both reviewers consider that the paper now deserves to be accepted, and I agree with them. One reviewer mentions some remaining comments to be taken care of. Therefore, I am pleased to inform you that your paper will be accepted for publication in Computo once the following minor points have been addressed:

- All points raised by reviewer v2H5 (pasted in the Comments below).

- Regarding the R code currently provided in the "functions" subdirectory, as pointed out by reviewer v2H5, it is not an R package. In order for other people to be able to run the proposed methods, we ask you to create a proper R package (and use that package in your article). This package can be minimal and simply hosted on your github if you do not wish to make a CRAN package.

- Another important point for publication: the figures should be improved. In particular, the figure labels and text are too small. Please use high-level standard plotting library such as ggplot2. This will greatly improve the aspect and readability of the paper.

---

> ### Decision · Editors_In_Chief · 2025-05-12
>
> **Approval:**
>
> I approve the AE's decision.
>
> **Comment To The Ae:**
>
> See the AE comments.

---

> ### Author Response · Authors · 2025-05-26
> **Minor revision**
>
> The requested minor revisions have been taken into account:
> - the discussion about the role of $X^O$ in the algorithm has been added
> - a R package poissonmix has been created, it is now used in the code of the paper, and its installable archive is avaibale in the Github repository (poissonmix_0.1.tar.gz)
> - all the figures are now done using ggplot2
> - typos has been fixed